# Establishing a Proteomics-Based Signature of AKR1C3-Related Genes for Predicting the Prognosis of Prostate Cancer

**DOI:** 10.3390/ijms24054513

**Published:** 2023-02-24

**Authors:** Xiaoli Cui, Changcheng Li, Jipeng Ding, Zhou Yao, Tianyu Zhao, Jiahui Guo, Yaru Wang, Jing Li

**Affiliations:** Department of Pharmacology, College of Basic Medical Sciences, Jilin University, Changchun 130021, China

**Keywords:** prostate cancer, AKR1C3, bioinformatics, label-free quantitative proteomics

## Abstract

Aldo-keto reductase family 1 member C3 (AKR1C3) plays an important role in prostate cancer (PCa) progression, particularly in castration-resistant prostate cancer (CRPC). It is necessary to establish a genetic signature associated with AKR1C3 that can be used to predict the prognosis of PCa patients and provide important information for clinical treatment decisions. AKR1C3-related genes were identified via label-free quantitative proteomics of the AKR1C3-overexpressing LNCaP cell line. A risk model was constructed through the analysis of clinical data, PPI, and Cox-selected risk genes. Cox regression analysis, Kaplan–Meier (K–M) curves, and receiver operating characteristic (ROC) curves were used to verify the accuracy of the model, and two external datasets were used to verify the reliability of the results. Subsequently, the tumor microenvironment and drug sensitivity were explored. Moreover, the roles of AKR1C3 in the progression of PCa were verified in LNCaP cells. MTT, colony formation, and EdU assays were conducted to explore cell proliferation and drug sensitivity to enzalutamide. Migration and invasion abilities were measured using wound-healing and transwell assays, and qPCR was used to assess the expression levels of AR target genes and EMT genes. CDC20, SRSF3, UQCRH, INCENP, TIMM10, TIMM13, POLR2L, and NDUFAB1 were identified as AKR1C3-associated risk genes. These risk genes, established using the prognostic model, can effectively predict the recurrence status, immune microenvironment, and drug sensitivity of PCa. Tumor-infiltrating lymphocytes and several immune checkpoints that promote cancer progression were higher in high-risk groups. Furthermore, there was a close correlation between the sensitivity of PCa patients to bicalutamide and docetaxel and the expression levels of the eight risk genes. Moreover, through in vitro experiments, Western blotting confirmed that AKR1C3 enhanced SRSF3, CDC20, and INCENP expression. We found that PCa cells with a high expression of AKR1C3 have high proliferation ability and high migration ability and were insensitive to enzalutamide. AKR1C3-associated genes had a significant role in the process of PCa, immune responses, and drug sensitivity and offer the potential for a novel model for prognostic prediction in PCa.

## 1. Introduction

It is estimated in the 2022 Cancer Statistics publication that PCa cases account for more than one-quarter of all new cancer cases and the second-highest number of cancer deaths [1]. The first-line treatment for PCa is androgen deprivation therapy (ADT; medical or surgical castration) to disrupt androgen receptor (AR) signaling [2]. However, tumors can recur and progress to fatal CRPC. This suggests that PCa cells can still grow adaptively in the absence of testis-derived androgens. In recent years, high expression of AKR1C3 has also been found in PCa. As a key member of the aldo-keto reductase superfamily, AKR1C3 is an enzyme mainly involved in the metabolism of steroid hormones [3,4]. It is closely related to a poor prognosis and the metastasis of tumors; in addition, AKR1C3 promotes the occurrence of drug resistance.

Based on the new theory of intratumoral androgen synthesis in PCa cells, AKR1C3 was found to play a pivotal role in the synthesis of testosterone and dihydrotestosterone, which are the most robust stimuli for the activation of the growth, proliferation, and metastasis of PCa cells [5,6]. PCa cells make use of androgens synthesized by themselves to adapt and progress to CRPC. A previous study identified AKR1C3 as a new critical regulator that can drive EMT through the activation of ERK signaling and the upregulation of a series of downstream transcription factors in PCa, whereas the attenuation of its expression or activity can reverse EMT and decrease cell invasion [7]. In addition, the upregulation of AKR1C3 was also observed in enzalutamide-resistant PCa cells. AKR1C3 promotes resistance to enzalutamide drugs by activating the androgen biosynthesis pathway and AR signal transduction. AKR1C3 not only plays a central role in endocrine androgen biosynthesis [4,8,9] but also is closely related to the expression of other genes. AKR1C3 reprograms AR/AR-V7 signaling in enzalutamide-resistant cells, induces AR-V7 overexpression, and stabilizes the AR-V7 protein in resistant cells through the alteration of the ubiquitin–proteasome system [10]. However, other AKR1C3-related genes have not been fully studied.

In this study, we found that AKR1C3 is a common gene throughout PCa progression, metastasis, and enzalutamide resistance. Furthermore, AKR1C3-related genes were identified through proteomics, providing important information for the study of AKR1C3-related molecules and signal pathways in PCa. Label-free proteomics is an analytical method without the need for protein labels; thus, it is more versatile and cheaper [11]. Moreover, we evaluated the correlation between our risk model and drug sensitivity. Finally, we verified the effect of AKR1C3 on cell proliferation, metastasis, and enzalutamide sensitivity in LNCaP cells in vitro, thus providing important information for the study of AKR1C3-related molecules and signal pathways in PCa.

## 2. Results

### 2.1. AKR1C3 as a Poor-Prognosis Gene in PCa Patients

The overall design of our study is shown in Figure 1. In PCa, poor-prognosis factors include castration-resistant, metastatic, and enzalutamide-resistant features, among others. We analyzed two associate castration-resistant datasets, GSE33316 and GSE39354. There were 161 common upregulated genes in the castration-resistant group (Appendix A). Two relevant metastatic datasets, GES6919 and GES7930, were analyzed. Both datasets shared 40 common upregulated genes in the metastatic group (Appendix A). Two correlated enzalutamide-resistant datasets, GSE151038 and GSE104935, were analyzed and found to include 243 co-upregulated genes (Appendix A). The database information is shown in Appendix A. Finally, we found only one gene (AKR1C3) that was present in all three castration-resistant, metastatic, and enzalutamide-resistant dataset analyses (Figure 2A). Subsequently, the role of AKR1C3 was analyzed using the data of 496 PCa patients from the TCGA database. The expression of AKR1C3 in PCa patients was not associated with age (Figure 2B). However, our results revealed that high T stage, N stage, and Gleason score values were associated with increased AKR1C3 expression (Figure 2C–E). Through overall survival (OS) and progress-free interval (PFI) event analysis, it was observed that AKR1C3 was more highly expressed in dead than alive patients (Figure 2F–G). Furthermore, Kaplan–Meier survival analysis was used to evaluate the significance of AKR1C3 expression in the disease-free survival of patients, as shown in Figure 2H. Survival analysis showed that PCa patients with a high expression of AKR1C3 had a short survival period. Furthermore, a receiver operating characteristic (ROC) curve was drawn, and the area under the curve (AUC) was 0.703 (Figure 2I). Therefore, the increased expression of AKR1C3 in patients was related to the progression of PCa, and AKR1C3 may be a risk factor for a poor prognosis among PCa patients.

### 2.2. PPI Network Construction and Risk Gene Selection

To evaluate the mechanism by which AKR1C3 leads to PCa progression, we constructed a stable-expression AKR1C3 cell line, LNCaP-AKR1C3. Compared with the LNCaP group, 1164 DEPs were identified in the LNCaP-AKR1C3 group (Appendix A), including 484 upregulated proteins and 680 downregulated proteins.

In order to investigate the interactions between AKR1C3-related proteins and uncover the core regulatory proteins among the 448 upregulated proteins, we constructed a PPI network using the STRING database. The network degree analysis for the node was conducted using the CytoNCA plugin. The proteins were identified on the basis of betweenness, closeness, degree, and eigenvectors greater than the median. After three filtration steps, 18 proteins were identified as hub genes (Figure 3A). The K–M plot demonstrated that the high-expression 18-signature group had unfavorable disease-free survival compared with the low-expression 18-signature group (Figure 3B). Eight risk proteins (CDC20, SRSF3, UQCRH, INCENP, TIMM10, TIMM13, POLR2L, and NDUFAB1) were identified using univariate Cox regression survival (hazard ratio > 1 and *p* < 0.05) (Figure 3C and Appendix A). The high AUC value in the ROC curve analysis showed that the eight risk genes had good distinguishing capabilities (Figure 3D). In the TCGA database, the expression of CDC20, UQCRH, TIMM10, TIMM13, POLR2L, and NDUFAB1 was upregulated in PCa tissue (Figure 3E). Further correlation analysis revealed that most of these eight risk genes were positively correlated with each other (Figure 3F).

### 2.3. Construction of Prognostic Risk Model for PCa

We used the expression and coefficients of these eight risk genes in TCGA to establish a risk scoring model, with the TCGA data as a training set. In the training set, the predictive performance of the genes was determined by univariate Cox regression. The total risk score was determined using the following equation:Riskscore = (3.28 × ExpCDC20) + (2.01 × ExpTIMM10) + (2.01 × ExpUQCRH) + (1.96 × ExpINCENP) + (1.93 × ExpNDUFAB1) + (1.82 × ExpSRSF3) + (1.68 × ExpTIMM13) + (1.52 × ExpPOLR2L).

Based on the median risk score value, the TCGA cohort was divided into low-risk and high-risk groups. The prognostic status of individuals between groups and the expression levels of the eight risk genes are presented in Figure 4A. The level of tumor recurrence in the high-risk score group was significantly higher compared with that in the low-risk score group (Figure 4B). The ROC curve analysis (Figure 4C) showed good discrimination, with AUCs of 0.63, 0.63, and 0.57 at 1-, 2-, and 5-year follow-ups, respectively.

### 2.4. Validation of Prognostic Risk Model for PCa

Two validation datasets, GSE116918 and GSE54460, were obtained from the Gene Expression Omnibus (GEO). Based on the median risk score validation, the datasets were divided into low-risk and high-risk groups. The distribution of the risk score recurrence status and the gene expression panel in the validation cohort are shown in Figure 5A,D. The Kaplan–Meier survival analysis showed that patients in the low-risk group had better BRC-free survival than the high-risk group (Figure 5B,E). The AUC values for 10, 8, and 5 years were 0.7, 0.66, and 0.62, respectively, in the GSE116918 dataset (Figure 5C). The AUC values for 10, 8, and 5 years were 0.65, 0.56, and 0.56, respectively, in the GSE54460 dataset (Figure 5F). The results on the validation set were similar to those on the training set, indicating that the model can effectively predict biochemical recurrence in PCa patients over the long term.

### 2.5. The Immune Cell Infiltration Analysis of the Risk Score Model in PCa

To further explore the relationship between the AKR1C3-related risk genes and the immune microenvironment, the TISIDB database was used to analyze the relationships between the risk genes and tumor-infiltrating immune cells. A total of 28 immune cell types and 8 risk gene relationships were identified (Figure 6A). The TCGA dataset of 498 patients was screened using CIBERSORTx to investigate the immune cell infiltration landscape, as shown in Figure 6B. The proportions of tumor-infiltrating immune cells were different between the low-risk group and high-risk group. The box plot shows that the levels of resting memory CD4 T cells, activated memory CD4 T cells, and M1 macrophages in the low-risk group were significantly higher than in the high-risk group; however, the levels of regulatory T cells, M0 macrophages, and M2 macrophages in the low-risk group were much lower than in the high-risk group (Figure 6C).

### 2.6. Correlation Analysis of the Risk Score Model with Immune Checkpoint Genes

Recent studies have shown that immune checkpoint inhibitors are an effective cancer treatment. We explored the correlations between the AKR1C3-related risk genes and immune checkpoint genes using TISIDB, as shown in Figure 7A. Surprisingly, the LAG3, ADORA2A, CTLA4, KDR, VTCN1, TIGIT, and CDC274 levels in the low-risk group were higher than those in the high-risk group (Figure 7B). Furthermore, we observed positive associations between AKR1C3 levels and the levels of multiple immune checkpoint genes, including LAG3, ADORA2A, CTLA4, KDR, VTCN1, TIGIT, and CDC274 (Figure 7C).

### 2.7. Drug Sensitivity Evaluation of AKR1C3 and Associated Risk Genes

In individualized cancer treatment, it is particularly important to use molecular biomarkers to predict the drug sensitivity of specific patients. Bicalutamide and enzalutamide are antiandrogen drugs used in the treatment of PCa. The correlations between the expression of AKR1C3-related risk signatures and the predicted drug reactions are shown in Figure 8A. AKR1C3, CDC20, INCENP, and SRSF3 were associated with decreased drug sensitivity to antiandrogens. According to the correlation analysis between AKR1C3-related risk signatures and drug sensitivity, it was found that there was a significant correlation between the risk signature expression and bicalutamide sensitivity. The higher the cor value, the stronger the correlation (Figure 8B). Our results also suggest that the high expression of AKR1C3-related risk signatures may be associated with greater sensitivity to chemotherapeutic drugs such as docetaxel (Figure 8C,D). Further, we used Western blotting to verify the expression of the AKR1C3-related bicalutamide-resistant proteins SRSF3, CDC20, and INCENP. As shown in Figure 8E, the expression levels of SRSF3, CDC20, and INCENP were upregulated in the AKR1C3-overexpressing group.

### 2.8. AKR1C3 Promotes PCa Cell Proliferation and Migration

In order to study the role of AKR1C3 in the malignant progression of PCa, firstly, we used Spearman correlation to analyze the correlation between AKR1C3 and EMT marker scores. As expected, in the TCGA dataset, the AKR1C3 expression score was positively correlated with the EMT characteristic score of PCa patients (Figure 9A). The results (Figure 9B) showed that AKR1C3 expression was high in 22RV1 cell lines and low in LNCaP cell lines. In the transwell assay and wound-healing assay (Figure 9C,D), AKR1C3 increased the cells’ migration ability. To verify whether AKR1C3 can regulate the expression of EMT-related markers in PCa, Western blotting was performed to detect EMT-associated genes. In LNCaP and 22RV1 cells, the overexpression of AKR1C3 can upregulate the expression of N-cadherin and Vimentin and downregulate the expression of E-cadherin (Figure 9E,G). qPCR was used to detect the expression levels of the EMT-related markers Vim, Twist, and Snail after the overexpression of AKR1C3 in LNCaP and 22RV1 cells. The results showed that AKR1C3 could significantly increase the expression of the EMT-related genes Vim, Twist, and Snail (Figure 9F,H). Furthermore, AKR1C3 knockdown reduced 22RV1 cell migration and the protein expression levels of EMT (Figure 9I,J). Together, these findings suggest that AKR1C3 may play an important role in the migration of LNCaP and 22RV1 cells.

### 2.9. Overexpression of AKR1C3 Decreased Enzalutamide Sensitivity in P Cells

Compared with the 10%-CSS group, the MTT assay showed that the overexpression of AKR1C3 increased LNCaP cell proliferation under androgen-deprived conditions (Figure 10A). The EdU incorporation results showed that the EdU content increased in the LNCaP-AKR1C3 group (Figure 10B). We noticed that, compared with the control group, the overexpression of AKR1C3 increased the expression of AR target genes such as PSA, FKBP5, TMRPSS2, and CENPN in LNCaP-AKR1C3 cells (Figure 10C).

To further evaluate whether AKR1C3 was related to enzalutamide sensitivity in PCa cells, the activity of LNCaP cells was significantly inhibited after treatment with 20 um enzalutamide for 48 h (Figure 10D). After the overexpression of AKR1C3, the sensitivity of cells to enzalutamide decreased (Figure 10E,H). However, in the 22RV1 cells insensitive to enzalutamide, 22RV1 cells recovered their sensitivity to enzalutamide after knocking down AKR1C3 (Figure 10F,G). Together, these findings demonstrate that AKR1C3 promoted PCa cell proliferation, which overcame enzalutamide-induced growth arrest.

## 3. Discussion

PCa is one of the most common malignant tumors threatening the health of men worldwide, especially metastatic castration-resistant prostate cancer (mCRPC). Patients are almost always resistant to endocrine therapy. Therefore, it is urgent to discover the mechanism of drug resistance against endocrine therapy and find biomarkers for a poor prognosis. Our previous research found that AKR1C3 promotes androgen synthesis in PCa and promotes PCa growth, and it was a promising biomarker for the progression of PCa. However, research on the nonenzymatic function of AKR1C3 remains limited, and the specific molecular mechanism of AKR1C3 in the processes of malignant progression of PCa, immune escape, and drug resistance has not been fully clarified. Therefore, the exploration of AKR1C3-related markers is crucial for the prognosis and personalized treatment of PCa.

In this study, we found that AKR1C3 may participate in the malignant progression of PCa by analyzing a PCa dataset. AKR1C3 had higher expression levels associated with a later T stage, higher lymph node metastasis rate, and higher Gleason score, as well as higher mortality. The adaptive overexpression of AKR1C3 involves the abnormal expression of many genes and the imbalance of related signal pathways in the progression of PCa. Proteomics plays an important role in the diagnosis, progression, and prognosis of PCa. According to the results of previous studies, proteins differentially expressed in PCa versus normal-appearing prostate tissue adjacent to the tumor, PCa versus benign prostatic hyperplasia, PCa versus benign prostatic hyperplasia, and PCa protein post-translational modification identify key proteins, as shown in Appendix A. In this study, we stably expressed AKR1C3 in LNCaP cells and carried out a label-free quantitative proteomic analysis to find AKR1C3-related proteins at the protein level. There were eight risk genes related to AKR1C3 that were screened through the PPI network, and we established a prognostic model based on these eight genes using the TCGA database as the training cohort. In the training cohort, a ROC curve analysis and K–M survival curve analysis showed that the model was a reliable prognostic model for differentiating between a high-risk group and a low-risk group. We used two different external GEO datasets to confirm the prognostic value of the eight risk genes. This confirmed that the eight-risk-gene model was reliable and stable in predicting the prognosis of PCa.

The prognostic model was composed of eight genes: CDC20, SRSF3, UQCRH, INCENP, TIMM10, TIMM13, POLR2L, and NDUFAB1. The gene expression profiles suggested that the expression levels of CDC20, UQCRH, TIMM10, TIMM13, POLR2L, and NDUFAB1 were upregulated in tumor tissue. They have been widely reported to be involved in the pathogenesis of a variety of tumors, including PCa. Cell division cycle 20 (CDC20) has a close relationship with PCa metastasis and may be a good predictor for mPCa [12]. CDC20 maintains the self-renewal ability of CD44+ PCa stem-like cells by promoting the nuclear translocation and transactivation of β-catenin. In addition, the knockdown of CDC20 can inhibit the expression of stemness-related genes and the self-renewal ability, chemoresistance, invasion capability, and tumorigenicity of CD44+ PCa stem-like cells [13]. The silencing of CDC20 suppresses mCRPC growth and enhances chemosensitivity to docetaxel by inhibiting Wnt/β-catenin signaling [14]. SRSF3 has been widely reported as an oncogene. The expression level of the splicing factor SRSF3 increases significantly in hypoxia. Mechanism studies have shown that hypoxia is an activator of SRSF3 in PC3 PCa cells [15]. In gastric cancer cells, the inhibition of SRSF3 alleviated proliferation and migration by regulating the PI3K/AKT/mTOR signaling pathway [16]. Ubiquinol cytochrome c reductase hinge (UQCRH) is a new protein that is located in the mitochondrial membrane and induces the production of mitochondrial reactive oxygen species (ROS). Its abnormally high expression might lead to the production of intracellular reactive oxygen species, thus promoting the expression of oncogenes and the occurrence and development of tumors. It is a novel prognostic factor for hepatocellular carcinoma and a potential diagnostic biomarker for lung adenocarcinoma [17,18]. The inner centromere protein (INCENP) is required for the correct mitotic localization of this kinase in cultured human cells. INCENP expression is elevated in a number of colorectal cell lines [19]. PRMT1 methylates arginine 887 on INCENP, and this methylation plays an important role in the interaction between INCENP and AURKB, which is a new mechanism for the regulation of chromosomal passenger complex function [20]. Translocase in the inner mitochondrial membrane 10 (TIMM 10) may play a key role in the evolution of myelodysplastic syndrome to acute leukemia [21]. TIMM13 is a translocation enzyme involved in introducing metabolite transporters from the cytoplasm into the mitochondrial inner membrane, and it is a direct target of miR-34a, which is involved in neuroblastoma, with poor clinical results [22]. The POLR2L gene encodes a subunit of RNA polymerase II, which was the main upregulated gene of drug resistance in gastric cancer found in a screening of the TCGA database [23]. The human mitochondrial acyl carrier protein NDUFAB1 is a nuclear-encoded subunit of complex I of the mitochondrial respiratory chain [24]. NDUFAB1 might be a potential target for lupus nephritis [25]. Although some genes have not been deeply studied in PCa, these eight risk genes can provide directions for further studies of the biological function and clinical characteristics of AKR1C3.

Based on the model of AKR1C3-related risk genes, we analyzed the differences between the high-risk and low-risk groups in the immune microenvironment, which provides a theoretical basis for searching for PCa treatment targets. The early changes in the normal tissue microenvironment can promote tumorigenesis, and, in turn, tumor cells can promote further tumor-promoting changes in the microenvironment. In China, more than 10 immune-checkpoint-inhibitor-related drugs have been listed, including PD-1/PD-L1 and CTLA-4 monoclonal antibodies. Some historic breakthroughs have been made in “hot tumors”, such as melanoma, lung cancer, and renal carcinoma [26,27,28]. Sipuleucel-T is the first immunotherapy drug approved by the FDA for patients with PCa. It can activate the patient’s own immune system to fight against tumor development and metastasis [29]. In so-called “cold tumor” PCa, there is a low tumor mutational burden (TMB) and an immunosuppressive tumor microenvironment (TME). Based on the AKR1C3-associated risk genes established in the risk model, the TME changed in the group with the high-risk index compared with the group with the low-risk index. Macrophages have complex roles in cancer [30]. M1 macrophages with cytotoxic potential are considered to be the “antitumor” phenotype, and M2 macrophages are usually considered to be the “tumor-promoting” phenotype. In PCa, the presence of tumor-associated macrophages (TAMs) is closely related to poor outcomes [31,32]. The tumor-associated macrophage count was higher in those with higher serum PSA, a higher Gleason score, or a later clinical T stage and those with PSA failure [33]. Tregs are a group of T cells that usually inhibit the immune system and cause tumor cells to escape immune control. The hyperactivity of Treg cells is directly related to a variety of diseases, such as chronic infections and cancer [34]. Furthermore, both the CTLA-4 pathway and regulatory T cells (Treg) are essential for the control of immune homeostasis [35]. The blockade of CTLA-4 and PD-1/PD-L1 in combination may, therefore, synergistically hinder Treg-mediated immune suppression, thereby effectively enhancing immune responses, including tumor immunity [36]. Notably, there was a positive correlation between AKR1C3-influenced high-risk models and the immune checkpoints CTLA4, CD274, LAG3, CTCM, KDR, and TIG2T. These results provide a theoretical basis for AKR1C3 as a therapeutic target for PCa. Furthermore, drug susceptibility analyses also indicated potential benefits from docetaxel for patients in the high-risk group.

Nevertheless, there are some limitations to this study. First, our research is based on proteomic results using the TCGA and GEO public databases, and this AKR1C3-related risk gene prediction model needs to be further verified in clinical practice. Secondly, our study only revealed the relationships between AKR1C3-related risk genes and patient prognosis, the tumor microenvironment, and drug sensitivity. The underlying mechanisms need to be further explored in experiments.

## 4. Materials and Methods

### 4.1. Data Acquisition and Preprocessing

The RNA-sequencing and clinical data of PCa patients were downloaded from The Cancer Genome Atlas (TCGA-PRAD) (https://portal.gdc.cancer.gov/, accessed on 23 March 2022). The GSE33316, GSE39354, GSE6919, GSE7930, GSE151038, GSE104935, GSE116918, and GSE4460 datasets were retrieved from the Gene Expression Omnibus (GEO) dataset (https://www.ncbi.nlm.nih.gov/geo/ accessed on 24 March 2022). The gene expression profiles were normalized using the scale method provided in the “limma” R package based on the set cutoff criteria of |log2FC| ≥ 1 and *p* < 0.05 for differentially expressed genes (DEGs).

Quantitative 4D label-free proteomic LNCaP-AKR1C3 samples and LNCaP samples were routinely cultured for 48 h, digested, and stored at −80 °C until the samples were ready for proteomic analysis. Proteomic analysis was conducted by Jingjie PTM Biolab Co., Ltd. (Hangzhou, China).

### 4.2. Construction of a Prognostic Signature Based on Risk Genes

We screened AKR1C3-related risk genes against the progression-free interval in patients with PCa using univariate Cox regression. *p* < 0.05 was considered significant for these genes. For the training group, the AKR1C3-based prognostic risk score was established by linearly combining the following formula with the expression-level-multiplied regression model: risk score = coef 1 × gene1 expression + coef 2 × gene 2 expression + coef 3 × gene 3 expression + coef N × gene N expression [37,38,39]. To evaluate the predictive power of the AKR1C3-based prognostic risk model, Kaplan–Meier curves and receiver operating characteristic (ROC) curves were used on the training and testing datasets. Univariate Cox proportional hazard regression and multivariate Cox proportional hazard regression analyses for OS were performed on the independent prognostic factor of the risk score.

### 4.3. Estimation of the Immune Microenvironment Composition

LM22 is a signature matrix file composed of 547 genes that can accurately distinguish 22 mature human hematopoietic populations. To estimate the relative proportions of these 22 infiltrated immune cell types in a tumor mass, the online analytical platform CIBERSORT (https://cibersortx.stanford.edu/ accessed on 23 April 2022) was used for each clinical sample of the high-risk and low-risk groups. The TISIDB (http://cis.hku.hk/TISIDB/ accessed on 23 April 2022) database was used to analyze the relationships between the risk gene expression levels and the tumor-infiltrating lymphocytes and immune checkpoints.

### 4.4. Drug Sensitivity Analysis

RNA-sequencing expression profiles and the corresponding clinical information for PCa samples were downloaded from the TCGA dataset (https://portal.gdc.com accessed on 23 March 2022). Predicted therapeutic drug responses for each sample were based on the DGSC database (The Genomics of Drug Sensitivity in Cancer, (https://www.cancerrxgene.org/ accessed on 6 April 2022). The “pRRophetic” software package in R software (3.6.1) was used to predict the half-maximum inhibitory concentration (IC50) of the therapeutic drugs [40]. The correlations between the target gene expression level and drug sensitivity were extracted and further studied using Spearman correlation analysis. All of the above analysis methods and R package were implemented using the R foundation for statistical computing (2020) version 4.0.3.

### 4.5. Cell Culture and Transduction

Cell lines were cultured in RPMI 1640 (Hyclone) medium with 10% fetal bovine serum (FBS, GIBCO), penicillin (100 U/mL), and streptomycin (100 μg/mL). LNCaP-AKR1C3 (LNCaP-AK) cells were generated by transient transfection of LNCaP cells with pCMV6 encoding AKR1C3. Before transfection, LNCaP cells were cultured in a medium without androgen (charcoal-stripped FBS, Biological Industries, Beit HaEmek, Israel) and phenol red for 24 h. Transfection was performed when the degree of cell fusion reached 40–50%. Before transfection, 10 μM androstanedione was added as a substrate. In the absence of androgen but with the substrate, transfection vector pCMV6 and AKR1C3 plasmids were treated with enzalutamide for 12 h.

### 4.6. MTT

Cell vitality was assessed using the 3-(4,5-dimethylthiazoleyl)-2,5-diphenyltetrazolium bromide (MTT) assay performed according to the manufacturer’s manual. LNCaP cells (2 × 10^3^ cells/well) were seeded in 96-well plates.

### 4.7. Colony-Forming Assay

In order to overexpress AKR1C3, LNCaP cells were plated in 6-well plates for 12 h and transfected with AKR1C3 plasmid. At 24 h after transfection, LNCaP and LNCaP-AKR1C3 cells were digested and seeded at 300 cells/well in 6-well plates. The fresh 1640 culture medium was replaced every 3 days, and colonies were counted 14 days after plating. The colonies were fixed with 4% paraformaldehyde at room temperature for 15 min and then stained with 0.01% crystal violet solution at room temperature for 30 min. The number of colonies was counted using the ImageJ software (Version 1.52).

### 4.8. Wound-Healing and Transwell Migration Assay

At 24 h after transfection, one day prior to setting up the transwell migration assay, the PCa cells were serum-starved overnight. Trypsinized cells were resuspended, and 6 × 10^4^ cells were seeded in 300 μL of the starvation medium in the upper compartment. At the same time, cells were seeded in 12-well plates at 1 × 10^5^ cells and scratched the next day. The analysis was carried out according to the manufacturer’s protocol [41].

### 4.9. Quantitative Real-Time PCR

The total RNA was extracted from the LNCaP and LNCaP-AKR1C3 cells with the RNAeasy™ Animal RNA Isolation Kit with Spin Column (Beyotime Biotechnology, Shanghai, China).

RNAs were reverse-transcribed into cDNAs using EasyScript Reverse Transcriptase (Transgen Biotech, Beijing, China). qRT-PCR was performed using FastStart Universal SYBR Green 2X Master Mix (Roche, Basel, Switzerland), and the primer sequences are listed in Appendix A.

### 4.10. Western Blot Analysis

In this experiment, whole-cell protein extracts were resolved on SDS-PAGE, and proteins were transferred to PVDF membranes. After blocking for 1 h at room temperature in 5% milk in TBS/0.1% Tween-20, membranes were incubated overnight at 4 °C with the following primary antibodies: Anti-Vimentin Rabbit mAb (PTM-5376, 1:1000, Jingjie PTM Biolabs, Hangzhou, China), Anti-E-Cadherin Rabbit mAb (PTM-5060, 1:1000, Jingjie PTM Biolabs), Anti-N-Cadherin Rabbit mAb (PTM-5221, 1:1000, Jingjie PTM Biolabs), and AKR1C3 antibody (A6229, 1:4000, Sigma-Aldrich, St. Louis, MO, USA). Then, they were incubated with secondary antibodies (ProteinTech, Wuhan, China) for 2 h (1:5000 dilution), followed by chemiluminescence detection.

### 4.11. Statistical Analysis

All statistical analyses were conducted using SPSS (Version 19.0). Student’s *t*-test and Spearman’s correlation analysis were performed to analyze the data. Statistical significance was defined as *p* < 0.05.

## 5. Conclusions

In conclusion, the signatures of AKR1C3-related genes effectively predicted the OS of PCa patients in the three databases and showed prognostic value in PCa patients. Thus, the model provides guidance for the treatment selection and prognosis of PCa. Based on the findings of this study, suitable therapeutic drugs can be selected for patients. The analysis of the expression of AKR1C3 and related proteins and the analysis of the tumor immune microenvironment provide ideas for immunotherapy and related drug research and development. In the future, we will conduct more in-depth research on the mechanism by which AKR1C3 leads to a decrease in androgen receptor blocker sensitivity.

## Figures and Tables

**Figure 1 ijms-24-04513-f001:**
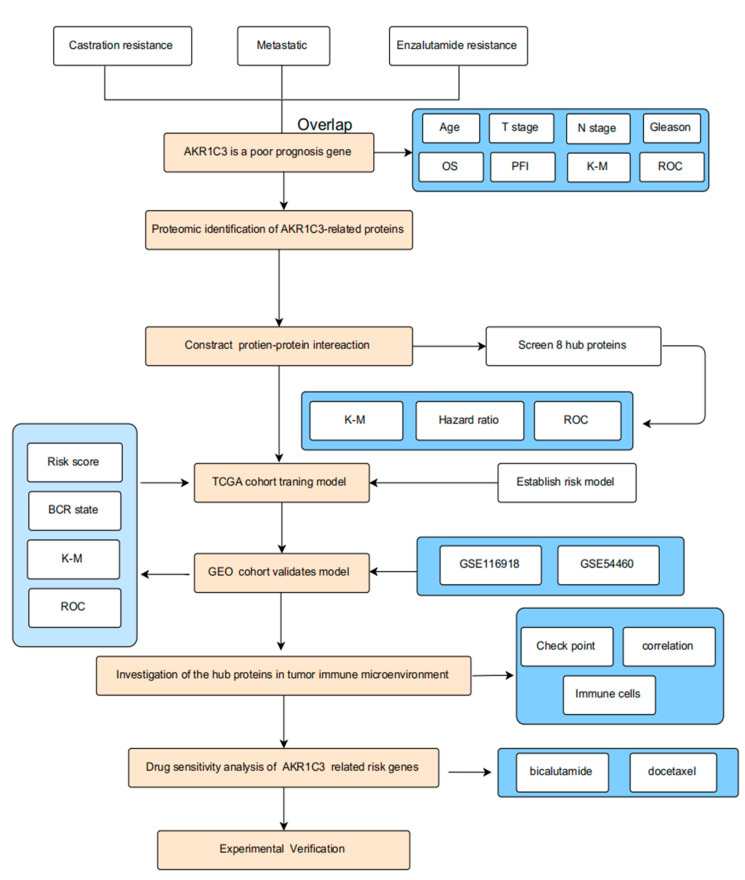
The flowchart of the present study design.

**Figure 2 ijms-24-04513-f002:**
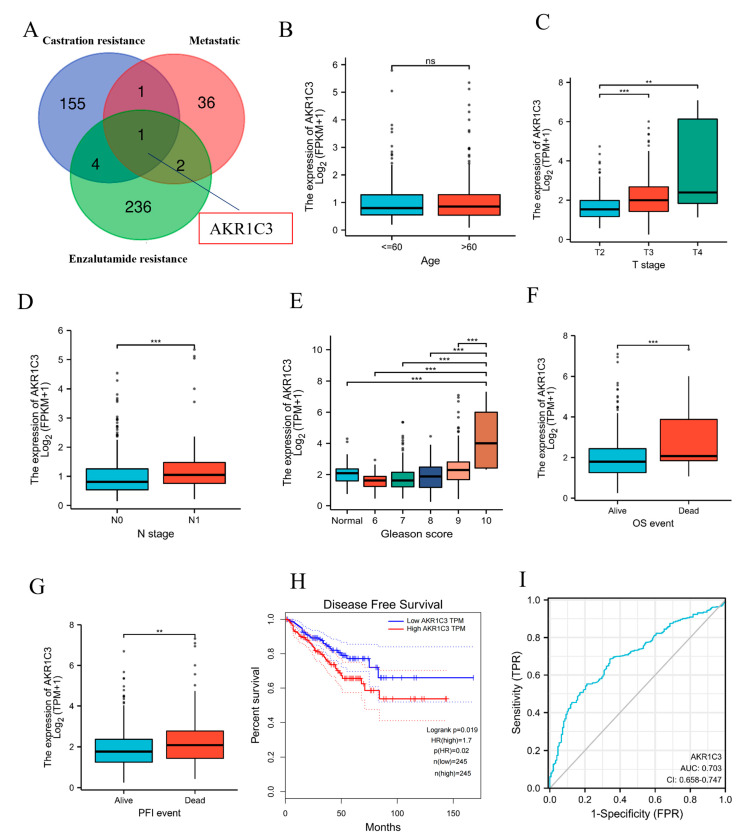
AKR1C3 is upregulated in human PCa patients and correlated with the poor prognosis of PCa patients. (**A**) AKR1C3 as candidate driver gene for PCa progression. Expression comparisons of AKR1C3 between different ages (**B**), T stages (**C**), N stages (**D**), and Gleason scores (**E**). AKR1C3 expression in alive and dead patients for OS events (**F**) and PFI events (**G**). (**H**) Kaplan–Meier disease-free survival curves for PCa patients in the high-AKR1C3-expression group and low-AKR1C3-expression group in the TCGA dataset. (**I**) ROC curves showed the predictive efficiency of AKR1C3 for patients in TCGA. ns.: no significance. ** *p* < 0.01, *** *p* < 0.001.

**Figure 3 ijms-24-04513-f003:**
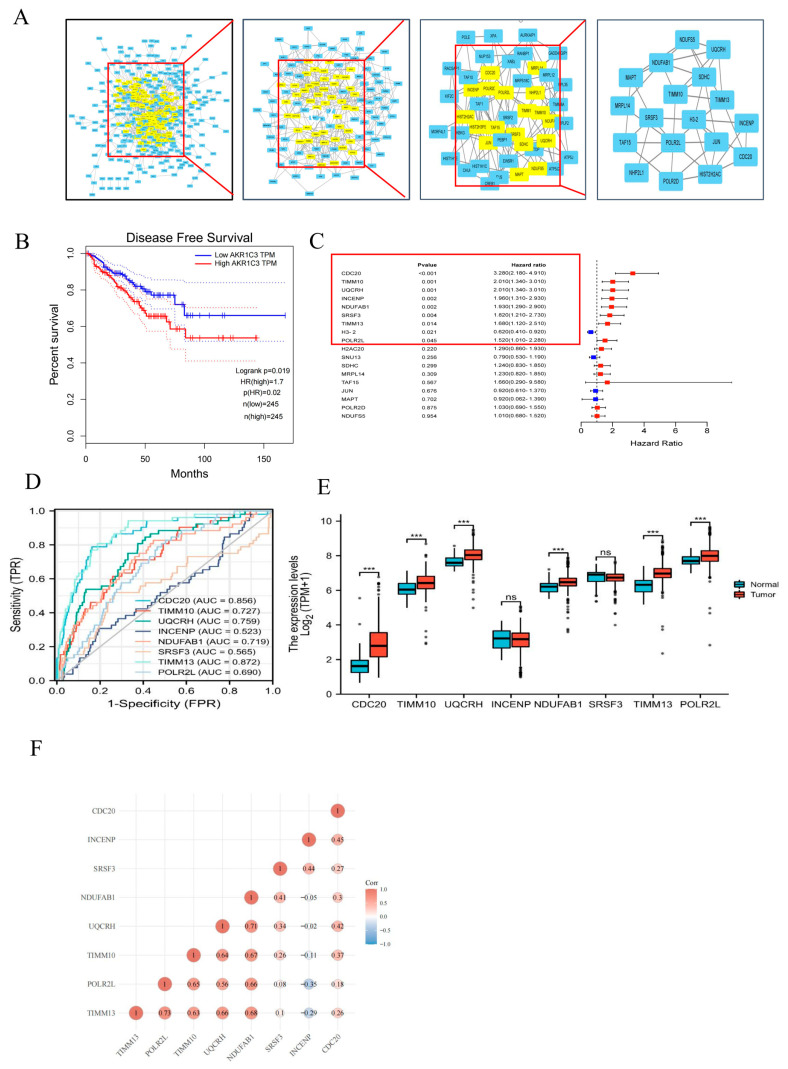
Screening of AKR1C3-related target proteins and their expression levels in PCa tissues. (**A**) A protein–protein interaction (PPI) network and hub module of differentially expressed proteins. (**B**) Kaplan–Meier analysis of the hub proteins. (**C**) Univariate Cox analysis regression analyses of the hub proteins. (**D**) ROC curves show the predictive efficiency of hub proteins for patients in the TGCA dataset. (**E**) The expression levels of hub proteins in normal and tumor tissues. (**F**) The correlations between each of the eight risk proteins in PCa. ns.: no significance. *** *p* < 0.001.

**Figure 4 ijms-24-04513-f004:**
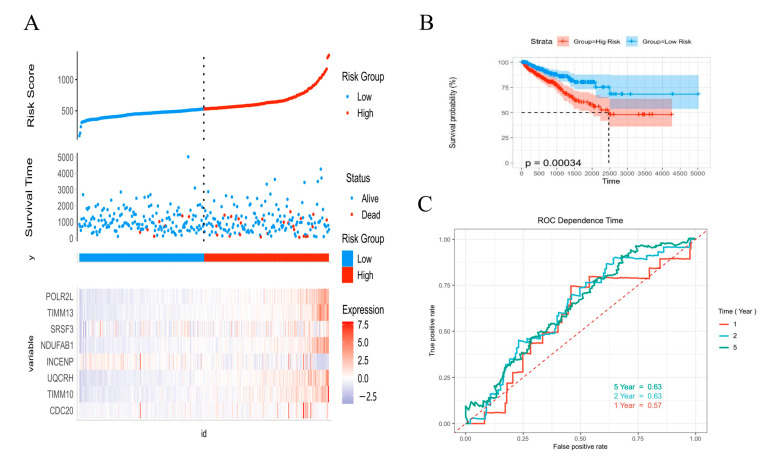
Evaluation of AKR1C3-associated proteins in the training cohort. (**A**) Risk curve and scatter plot for the risk score and survival status of each patient based on the risk score. (**B**) Kaplan-Meier analysis for the prognosis prediction of the risk score model. (**C**) Time-dependent ROC curves for the prognostic prediction of the risk score model at 1, 2, and 5 years.

**Figure 5 ijms-24-04513-f005:**
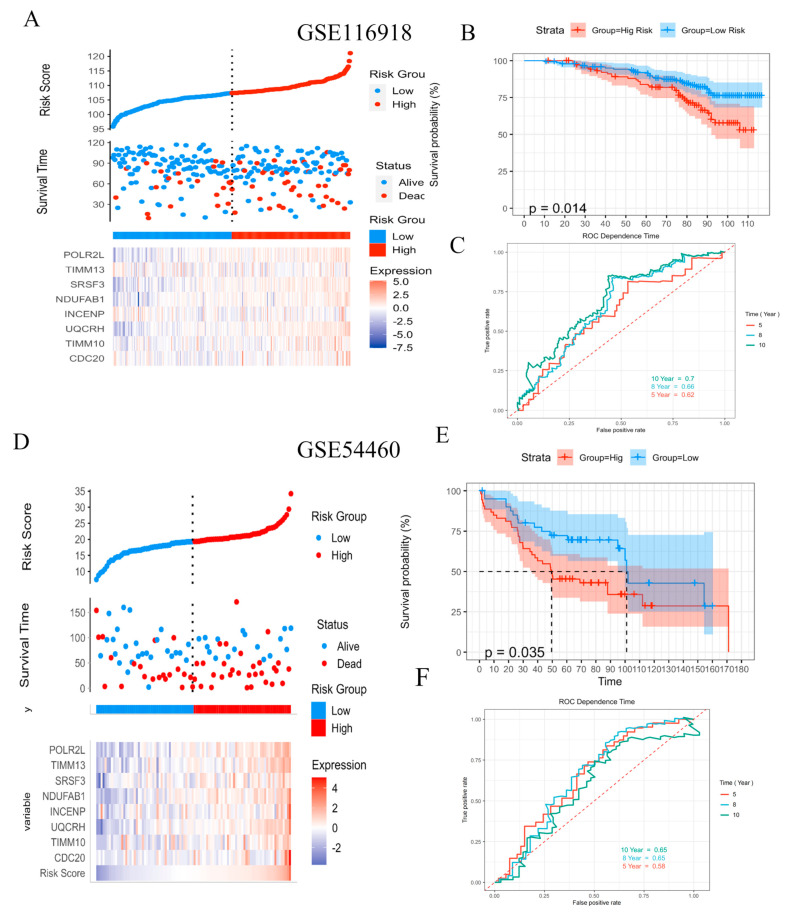
Evaluation of AKR1C3-associated proteins in the test cohort. (**A**,**D**) Risk curve and scatter plot for the risk score and survival status of each patient based on the risk score. (**B**,**E**) Kaplan-Meier analysis for the prognostic prediction of the risk score model. (**C**,**F**) Time-dependent ROC curves for the prognostic prediction of the risk score model at 5, 8, and 10 years, respectively.

**Figure 6 ijms-24-04513-f006:**
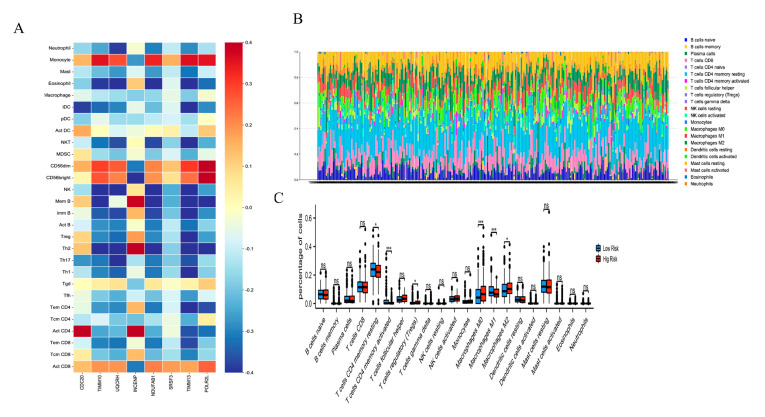
(**A**) The correlation between ARK1C3-associated proteins and tumor-infiltrating lymphocytes across PCa. (**B**) Landscape of immune cell infiltration. (**C**) Different proportions of tumor-infiltrating cells between high-risk and low-risk groups. ns.: no significance. * *p* < 0.05, *** *p* < 0.001.

**Figure 7 ijms-24-04513-f007:**
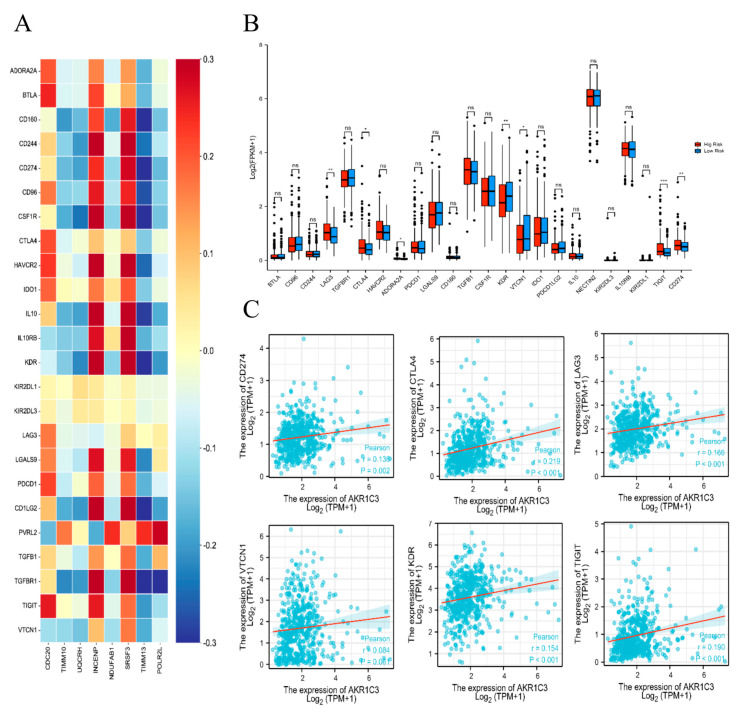
(**A**) The correlation between AKR1C3−associated risk genes and immunoinhibitory factors across PCa. (**B**) The expression levels of immune checkpoint molecules in high-risk and low-risk groups. (**C**) The correlation between AKR1C3 and immune checkpoint molecules. ns.: no significance. * *p* < 0.05, ** *p* < 0.01, *** *p* < 0.001.

**Figure 8 ijms-24-04513-f008:**
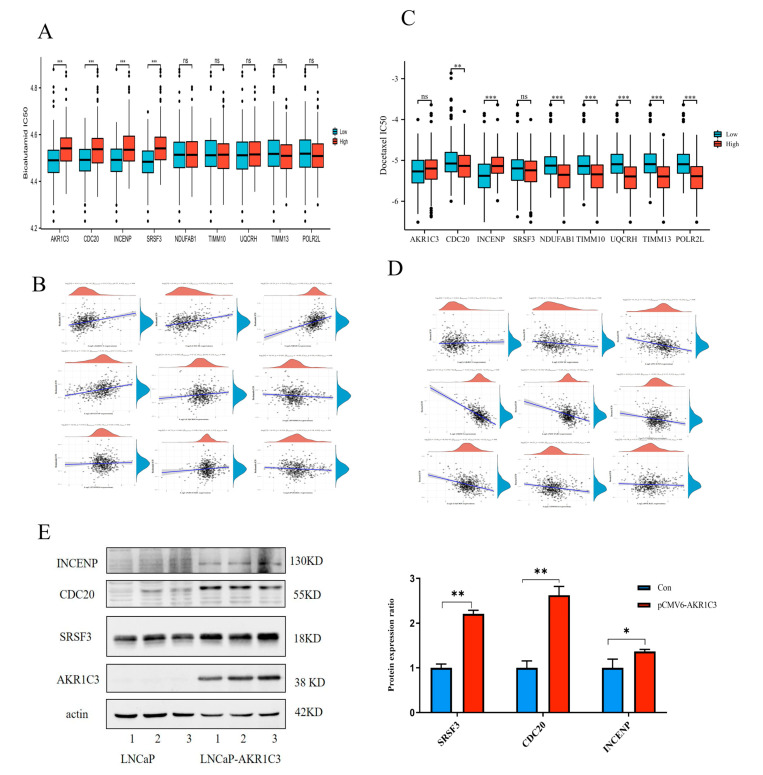
Drug sensitivity analysis of AKR1C3-related risk genes. (**A**) The distribution of bicalutamide IC50 score and docetaxel (**C**) IC50 score. (**B**) Spearman correlation analysis between bicalutamide (**D**) docetaxel IC50 score and AKR1C3-related gene expression. (**E**) Western blot verified the expression of drug resistant proteins SRSF3, CDC20 and INCENP in LNCaP-AKR1C3 cells. ns.: no significance. * *p* < 0.05, ** *p* < 0.01, *** *p* < 0.001.

**Figure 9 ijms-24-04513-f009:**
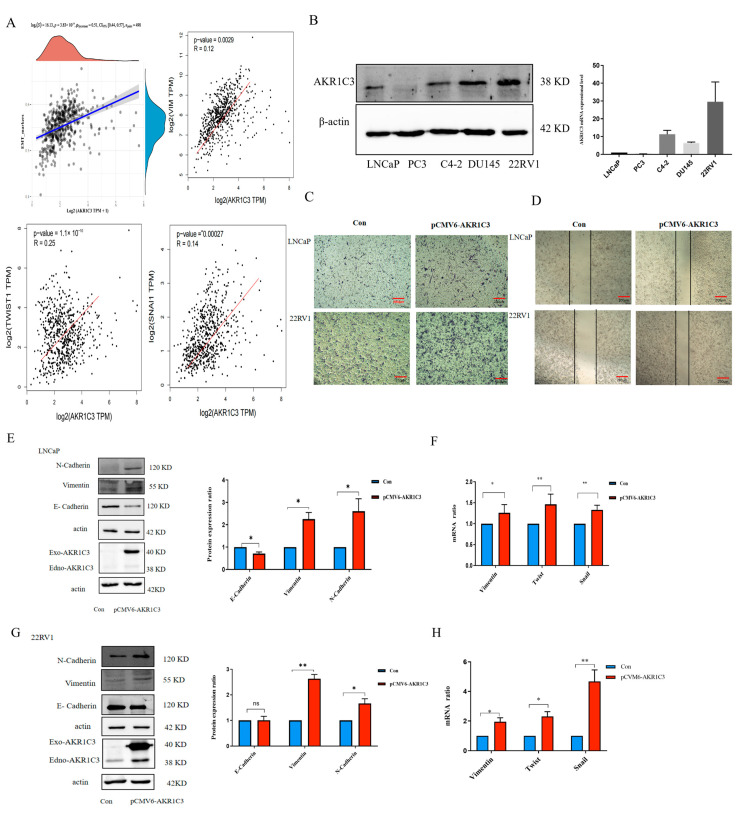
AKR1C3 promotes PCa cell EMT. (**A**) Spearman correlation analysis was used to evaluate the relationship between AKR1C3 and EMT marker protein levels. (**B**) AKR1C3 protein and mRNA expression levels in various PCa cell lines. Transwell migration assay (**C**) and wound-healing assay (**D**) were performed in LNCaP cells and 22RV1 cells. (**E**,**G**,**J**) Western blotting detected EMT-associated markers in LNCaP cells and 22RV1 cells. (**F**,**H**) Changes in marker expression were analyzed using qPCR. (**I**) Transwell migration assay and wound-healing assay at 48 h in AKR1C3 knockdown 22RV1 cells. Data are presented as mean ± SD and are representative of three independent experiments. ns.: no significance.* *p* < 0.05, ** *p* < 0.01. Scale bar: 200 um.

**Figure 10 ijms-24-04513-f010:**
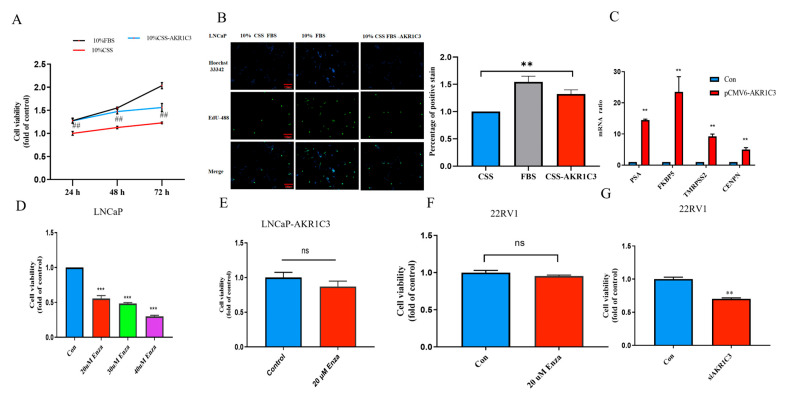
AKR1C3 promoted cell proliferation and increased enzalutamide resistance in LNCaP cells. (**A**) The viability of AKR1C3 overexpression in LNCaP cells was measured using the MTT assay. (**B**) EdU-staining analysis of cell proliferation. (**C**) qPCR analysis of expression of AR target genes. (**D**) Viability of LNCaP cells after being treated with enzalutamide. (**E**) The viability of AKR1C3 overexpression in LNCaP cells was measured using the MTT assay after treatment with 20 uM enzalutamide and 10% CSS-FBS. (**F**) Viability of 22RV1 cells after being treated with 20 uM enzalutamide for 48 h. (**G**) The MTT assay was used to determine the activity of enzalutamide in 22RV1 cells after knocking down AKR1C3. (**H**) Clone formation assays in AKR1C3 overexpression cells treated with enzalutamide. ns.: no significance. ** *p* < 0.01, ^##^
*p* < 0.01, *** *p* < 0.001. Scale bar: 200 um.

## Data Availability

Publicly available datasets were analyzed in this study. These data can be found here: (https://portal.gdc.cancer.gov/), (https://www.ncbi.nlm.nih.gov/geo/), (http://cis.hku.hk/TISIDB/), (https://www.cancerrxgene.org/), accessed on 23 March 2022 to 30 April 2022.

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
