# Peer review of "Establishing a Proteomics-Based Signature of AKR1C3-Related Genes for Predicting the Prognosis of Prostate Cancer"

_ijms, 2023, doi:10.3390/ijms24054513_

Round 1
Reviewer 1 Report
“Base on proteomics established signature of AKR1C3 relative genes to predict the prognosis of Prostate Cancer” by Xiaoli Cui et al describes the potential and the essential of AKR1C3-related genes to predict the response to chemotherapeutic drugs and a prognosis factor in prostate cancer. The manuscript is generally well-written. I have several issues that need to be clarified before publication.
1. The authors combine three different type datasets of prostate cancers including castration-resistant (GSE33316 and GSE39354), metastatic (GES6919 and GES7930), enzalutamide-resistant (GSE151038 and GSE104935). The authors should describe the clinical pathological features because metastatic prostate cancer patients may also have been castration-resistant.
2. The authors constructed a stable express AKR1C3 cell line (LNCaP-AKR1C3). But there is no included figure showing the success of this experience.
3. The authors calculated the risk score “Risk score=coef 1*gene1 expression+ coef 2*gene 2 expression+ coef 3* gene 3 expression +coef N* gene N expression”. Please clarify this formula, is this common use in bioinformatics research?
4. There are grammatical errors and wording. The manuscript needs to be thoroughly revised.
Author Response
- Comment: The authors combine three different type datasets of prostate cancers including castration-resistant (GSE33316 and GSE39354), metastatic (GES6919 and GES7930), enzalutamide-resistant (GSE151038 and GSE104935). The authors should describe the clinical pathological features because metastatic prostate cancer patients may also have been castration-resistant.
Response: These datasets are not all clinical samples, but also cell line samples. We describe the information of each data set as follows.
GSE33316: Compare the gene expression of 5 LuCaP35 xenografts from non-treated mice (Control), and 5 androgen-deprived LuCaP35 xenografts from castrated mice (Castration).human prostate xenograft tumor LuCaP35.
GSE39354: The different expressed gene in human prostate VCaP and LNCaP cancer cells. Vcap Cells was castration-resistant cells and LNCap cells was not castration-resistant cells.
GES6919: Normal prostate tissue free of any pathological alteration from organ donor and Metastatic prostate tumor samples in para tracheal lymph node from patient.
GSE7930:The different gene expressed between metastasis PC3 and normal cell.
GSE151083:C42B and C42B-enzalutamide resistant cells.
GSE104935: MDV3100 is an androgen receptor antagonist. This data set shows the difference of gene expression between C4-2B and C4-2B MDV3100resistant cells.
- Comment: The authors constructed a stable express AKR1C3 cell line (LNCaP-AKR1C3). But there is no included figure showing the success of this experience.
Response: Thanks for the question raised by the reviewer, we add the protein expression of AKR1C3 to Figure 8E.
- Comment: The authors calculated the risk score “Risk score=coef 1*gene1 expression+ coef 2*gene 2 expression+ coef 3* gene 3 expression +coef N* gene N expression”. Please clarify this formula, is this common use in bioinformatics research?
Response: This is common use in bioinformatics research, There are no less than 200 articles using this method for bioinformatics analysis, and we will quote the representative articles using this method in the manuscript.
- Comment: There are grammatical errors and wording. The manuscript needs to be thoroughly revised.
Response: The manuscript has undergone English language editing by MDPI and we have confirmed the revised manuscript.
Please see the attachment for the revised manuscript

Reviewer 2 Report
The study tried to prove that the signatures of AKR1C3 relative genes can effectively predict the OS of PCa patients and has prognostic value in PCa patients.
The content of bioinformatics analysis is relatively detailed, but there are many defects in the content of experimental verification.
1. Oncology experiments require at least two cell lines.
2. Western Blot is required to detect the expression of EMT and AKR1C3 related genes.
3. The transwell image in Figure 9B is not clear.
4. The Figure 10B image does not reflect statistical differences.
5. Please explain why the cloning experiment can be carried out by transient transfection.
Author Response
- Comment:1. Oncology experiments require at least two cell lines.
Response: Thank you for pointing this out, about AKR1C3 promote prostate cancer cells EMT, promoted cell proliferation and increase enzalutamide resistance, We used another prostate cancer cell line 22RV1 for verification, Results added to Figure 9(B-G) and Figure (F,G)
- Comment: Western Blot is required to detect the expression of EMT and AKR1C3 related genes.
Response: Thank you for your suggestions, We have used western blot to detect the expression of EMT-related protein E -cadherin, N- cadherin and V in LNCaP cells and 22RV1 cells.
3.Comment: The transwell image in Figure 9B is not clear.
Response: Thank you for pointing this out. In the newly revised manuscript, we have replaced the picture, as shown in Figure 9B.
- Comment: The Figure 10B image does not reflect statistical differences.
Response: We have carried out statistical analysis on Figure 10B, and the analysis results are on the right side of Figure 10B.
- Comment: Please explain why the cloning experiment can be carried out by transient transfection.
Response: In our experiment, AKR1C3 was transient transfected into cells, and it was found that overexpression of AKR1C3 could promote the ability of clone formation and reduce the sensitivity to enzalutmide. Although there was plasmid loss in transient transfection, our experimental results showed that the role of AKR1C3 in the malignant progression of prostate cancer cells was very clear even under the condition of transient transfection.

Reviewer 3 Report
In this manuscript, Cui et al. investigate the role of AKR1C3 and its related genes in PC based on proteomics, highlighting the importance of AKR1C3 and its related genes in process of PCa, immune responses, drug sensitivity. The findings are important, however, the story of each part end suddenly, and more validation experiments are required to present a comprehensive study.
Major points:
1. The key findings should be validated in other PC cell lines.
2. The author should link the AKR1C3 to its related genes in different biological functions. The role of distinctive AKR1C3-related genes/proteins analyzed by proteomics data should be further validated by using specific biological experiments with or without AKR1C3 expression.
Minor points:
1. The language should be double-checked by a native speaker.
2. Check the format errors.
Author Response
1.Comment:1. The key findings should be validated in other PC cell lines.
Response: Thank you for pointing this out, about AKR1C3 promote prostate cancer cells EMT, promoted cell proliferation and increase enzalutamide resistance, We used another prostate cancer cell line 22RV1 for verification, Results added to Figure 8(B-G) and Figure (F,G).
- Comment: The author should link the AKR1C3 to its related genes in different biological functions. The role of distinctive AKR1C3-related genes/proteins analyzed by proteomics data should be further validated by using specific biological experiments with or without AKR1C3 expression.
Response: Thank you very much for the comments. Through bioinformatics analysis, it was found that AKR1C3 related proteins SRSF3, CDC20 and INCENP were positively correlated with the resistance of androgen antagonist bicalutamide. We used western blot to verify the expression of overexpressed AKR1C3 on drug-resistant proteins SRSF3, CDC20 and INCENP. The results are supplemented to Figure 8E.
3.Comment:The language should be double-checked by a native speaker.
Response: The manuscript has been modified using the recommended native speaker company, and we have confirmed the revised manuscript.
4.Comment:Check the format errors.
Response: We appreciate this helpful suggestion. We have changed the format in the new manuscript.
Reviewer 4 Report
This work deals with LF-proteomic analysis of AKR1C3 in prostate cancer samples (PCa).
The associated genes of AKR1C3 were analysed using proteomic approach in LNCaP cell line.
From the results, it has been reported that the risk genes established ability to build up the prognosis model to predict the recurrence status, immune u-environment and drug sensitivity related to PCa.
Remarks:
Please, follow the down below upgrade in the last paragraph of introduction:
“Label-free proteomics is an analytical method neglecting the need for the protein labels, thus it is more versatile and cheaper [https://doi.org/10.1515/biol-2019-0070]. In this study, a label-free proteomic approach was used and we found a common gene AKR1C3 with PCa progression, metastasis and enzalutamide resistance and identify AKR1C3 related genes, providing important information for the study of AKR1C3 related molecules and signal pathways in PCa.
Please, make a table of the PCa related genes analysed by proteomic analysis till now in already published genes and also include genes analysed by this study design in the table.
In the conclusion, please give some future aims of the authors about what can be done in this research field.
Author Response
1.Comment: Please, follow the down below upgrade in the last paragraph of introduction:
“Label-free proteomics is an analytical method neglecting the need for the protein labels, thus it is more versatile and cheaper [https://doi.org/10.1515/biol-2019-0070]. In this study, a label-free proteomic approach was used and we found a common gene AKR1C3 with PCa progression, metastasis and enzalutamide resistance and identify AKR1C3 related genes, providing important information for the study of AKR1C3 related molecules and signal pathways in PCa.
Response: Thank you very much for the comments. I have revised the introduction of the manuscript according to the comments.
- Comment: Please, make a table of the PCa related genes analysed by proteomic analysis till now in already published genes and also include genes analysed by this study design in the table.
Response: Through data review, the application of proteomics in prostate cancer is very valuable, such as PCa versus normal appearing prostate tissue adjacent to the tumor, PCa versus benign prostatic hyperplasia, PCa versus benign prostatic hyperplasia and prostate cancer protein post-translational modification by proteomics, The above description was presented in the discussion and a table was made, which is shown in Supplementary Table 3.However, we did not find the protein consistent with that in this manuscript.
3.Comment:In the conclusion, please give some future aims of the authors about what can be done in this research field.
Response: Thank you for your comments. According to this study, the prognosis of patients can be predicted according to the expression of AKR1C3 and related proteins. Based on the findings of this study, suitable therapeutic drugs can be selected for patients. Immunotherapy plays an important role in the treatment of tumors. Analysis of the expression of AKR1C3 and related proteins, and analysis of tumor immune microenvironment provide ideas for immunotherapy and related drug research and development. We made changes in the the revised manuscript.

Round 2
Reviewer 3 Report
The authors have addressed some of my concerns. However, the following concerns should be figured out.
1. It seems that the LNCap cells does not express AKR1C3. What is the rational to choose this cell line? The author should compare AKR1C3 expression in multiple PC cell lines.
2. The loss-of-function part are missing, the author should use AKR1C3-high PC cell to knockdown AKR1C3.
3. Please check the figures. Figure 8(B-G) and Figure (F,G)?
Author Response
1.Comment:1. It seems that the LNCap cells does not express AKR1C3. What is the rational to choose this cell line? The author should compare AKR1C3 expression in multiple PC cell lines.
Response: Thank you for pointing this out. We previously observed the expression of AKR1C3 in different prostate cancer cell lines through literature and western blot. In AR-positive cell lines, LNCaP cells are androgen-sensitive prostate cancer cells with low expression of AKR1C3. Therefore, the effect of AKR1C3 was observed by overexpressing AKR1C3 in LNCaP cells. In the new manuscript, we detected the expression of AKR1C3 in different prostate cancer cell lines. See Figure 9B.
- Comment: The loss-of-function part are missing, the author should use AKR1C3-high PC cell to knockdown AKR1C3.
Response: Thank you very much for the comments. In the loss-of-function experiment, we selected 22RV1 cells with high expression of AKR1C3. The results were shown in Figure 9I, J and 10F.
3.Comment:Please check the figures. Figure 8(B-G) and Figure (F,G)?
Response: Thank you very much for the comments. In the last reply to you, because of my mistake, I wrote Fig9 into Fig8.This section has been changed in the revised manuscript.

Reviewer 4 Report
Authors have reacted to given queries.
Author Response
Thank you very much for the comments.
Round 3
Reviewer 3 Report
Regarding the loss-of-function, to exclude the nonspecificity the author should use at least 2 siRNAs. In addition, the knockdown efficiency is missing, as well as the overexpression part.
Author Response
- Comment: Regarding the loss-of-function, to exclude the nonspecificity the author should use at least 2 siRNAs. In addition, the knockdown efficiency is missing, as well as the overexpression part.
Response: We appreciate your comments and suggestions for improving our manuscript. In overexpression part, the expression of AKR1C3 protein transfected with AKR1C3 plasmid in LNCaP and 22RV1 cells is shown in Fig 9E and 9G.For the loss-of-function part, in the previous work[Cui, X.; Yao, Z.; Zhao, T.; Guo, J.; Ding, J.; Zhang, S.; Liang, Z.; Wei, Z.; Zoa, A.; Tian, Y.; et al. siAKR1C3@PPA complex nucleic acid nanoparticles inhibit castration-resistant prostate cancer in vitro. Front. Oncol. 2022, 12, 1069033, doi: 10.3389/fonc.2022.1069033.], we screened 2 effective siRNA sequences. The results were shown in Figure 9 J .